# Low temperature near-field fingerprint spectroscopy of 2D electron systems in oxide heterostructures and beyond

Julian Barnett [1], Konstantin G. Wirth[1], Richard Hentrich[2], Yasin C. Durmaz[2,3], Marc-André Rose[4], Felix Gunkel [4] & Thomas Taubner [1] ✉

Confined electron systems, such as 2D electron gases (2DEGs), 2D materials, or topological insulators, show great technological promise but their susceptibility to defects often results in nanoscale inhomogeneities with unclear origins. Scattering-type scanning near-field optical microscopy (s-SNOM) is useful to investigate buried confined electron systems non-destructively with nanoscale resolution, however, a clear separation of carrier concentration and mobility was so far impossible in s-SNOM. Here, we predict a previously inaccessible characteristic "fingerprint" response of the prototypical $LaAlO_3$/$SrTiO_3$ 2DEG, and verify it using a state-of-the-art tunable narrow-band laser in mid-infrared cryo-s-SNOM at 8 K. Our modeling allows us to separate the influence of carrier concentration and mobility on fingerprint spectra and to characterize 2DEG inhomogeneities on the nanoscale. Finally, we model the surface accumulation layer in doped InAs, to show that our fingerprint spectra are a universal feature and generally applicable to confined electron systems, like topological insulators or stacked van-der-Waals materials.

Layered heterostructures of complex oxides have become an integral part of novel electronic, spintronic, and magneto-ionic device concepts[1–4]. Such structures often contain high local densities of electronic charge carriers, giving rise to correlation phenomena and emerging properties, such as metallic behavior[5–7], superconductivity[8–10], and magnetism[11–13], not found in the adjacent bulk materials. This poses new challenges to characterize high carrier concentrations in spatially confined and buried electronic systems, especially at cryogenic temperatures. Scattering-type scanning near-field optical microscopy (s-SNOM)[14,15] is a non-destructive method that uses strong optical near-fields and exhibits high surface sensitivity, sub-surface capabilities, and applicability in a broad spectral range. This makes s-SNOM an ideal candidate to investigate highly-confined electron systems that exist in the transdimensional regime[16] between purely 2D and bulk 3D, such as van-der-Waals materials (e.g., few-layer graphene[17]) or oxide heterostructures (e.g., $LaAlO_3/SrTiO_3$[5]). However,

the spectral information that is observed in s-SNOM is convoluted in terms of probing depth and in terms of different physical excitations[15], such as phonons, plasmons, or interband transitions. This imposes challenges on the qualitative and quantitative understanding of the acquired s-SNOM data, especially in layered structures. For one, the near-field response consists of contributions of multiple layers simultaneously, which makes a detailed understanding and modeling of the near-field response a crucial requirement. Moreover, this calls for the identification of dedicated spectral regions that provide characteristic information on the layer of interest ("fingerprint" regions), adding requirements to the accessible frequency range of light sources at a sufficient signal-to-noise ratio.

The 2D electron gas (2DEG) at the interface between the two insulators $LaAlO_3$ (LAO) and $SrTiO_3$ (STO)[5] has been studied extensively as a model system for high-concentration correlated electron systems. However, the local formation process and the influence of

[1]I. Institute of Physics (IA), RWTH Aachen University, Aachen, Germany. [2]attocube systems GmbH, Haar, Germany. [3]Department of Physics, Ludwig Maximilians University of Munich, Munich, Germany. [4]Peter Grünberg Institute (PGI-7) and Jülich-Aachen Research Alliance (JARA-FIT), Forschungszentrum Jülich, Jülich, Germany. ✉e-mail: taubner@physik.rwth-aachen.de

defects on the electronic properties are still not fully understood[18–20]. s-SNOM was shown to be sensitive to the 2DEG in LAO/STO[21] and even allow in principle, for the extraction of local electronic properties with nanoscale lateral resolution[22–24]. Due to the metallic behavior of the system, the 2DEG mobility increases at low temperatures[5], which should lead to a stronger sensitivity of s-SNOM to the 2DEG properties at cryogenic temperatures (cryo-s-SNOM) compared to room temperature. Previous s-SNOM studies were limited to indirectly probing the 2DEG via secondary effects, i.e., damping of the phonon near-field resonance of the STO substrate[23] or as a constant background at higher frequencies[22,24]. During these studies, direct mapping of the local charge carrier density turned out to be difficult, as the influence of different parameters, e.g., carrier concentration and mobility, can compensate, resulting in similar near-field spectra[24].

In this work, we predict the existence of a spectral fingerprint region, where a characteristic scattering response of the 2DEG can be obtained directly, resulting in a near-field spectroscopic method that allows for the separation of carrier concentration and mobility. This spectral region was previously inaccessible for s-SNOM on LAO/STO due to missing light sources with sufficient signal-to-noise ratio (cf. Supplementary Information S1), as the overall scattering signal of the fingerprint region is very low. Here, we use a newly-developed tunable narrow-band mid-infrared laser to investigate LAO/STO in the 2DEG fingerprint region, using cryo-s-SNOM at 8 K. This allows us to use a normalization procedure that is highly sensitive to the influence of the 2DEG (cf. Supplementary Information S2) and can be used to characterize the local electronic properties in detail.

## Results

Figure 1a presents a simplified view of the experimental setup (cf. Methods for details), with the metal-coated AFM tip scanning across the LAO/STO sample while oscillating with tapping frequency $\Omega_{tip}$ and being illuminated with a focused laser beam. The measurements presented in this work were recorded with a cryo-neaSCOPE by attocube

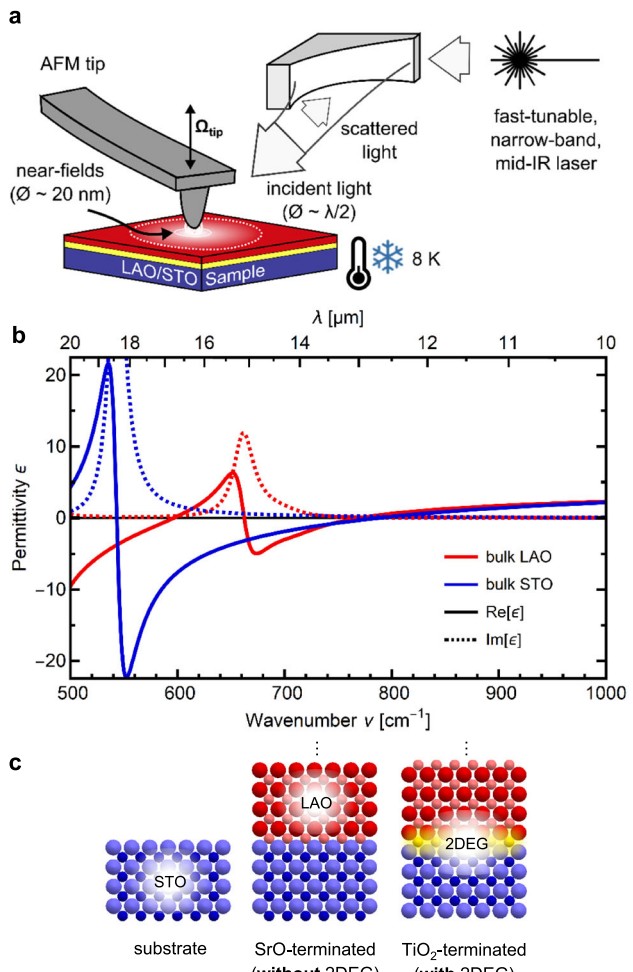

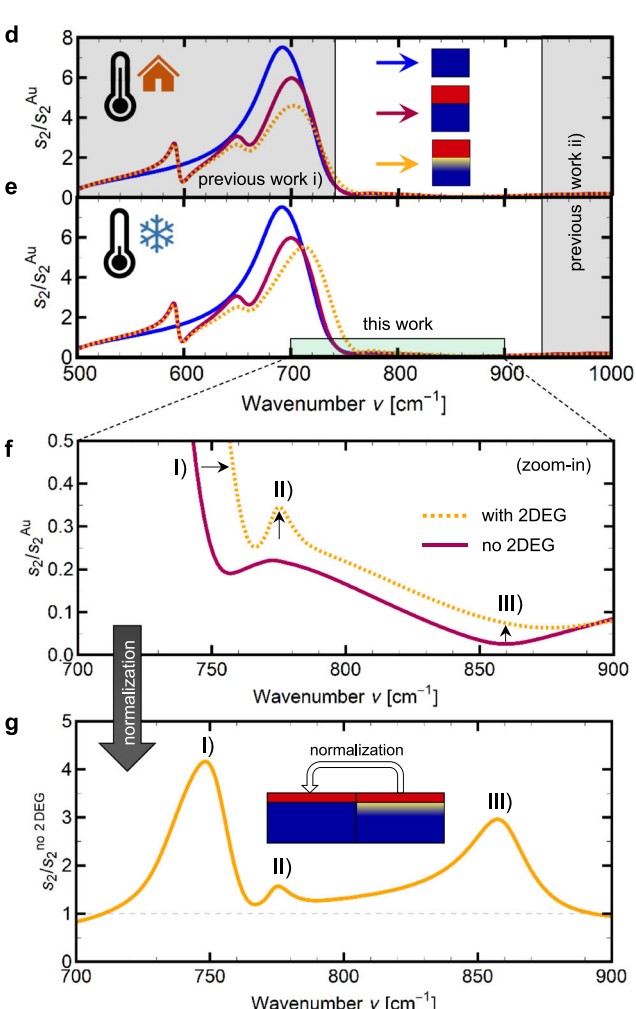

**Fig. 1 | Investigation of LaAlO$_3$/SrTiO$_3$ (LAO/STO) interfaces using scanning near-field optical microscopy (s-SNOM) and spectroscopy in the 2D electron gas (2DEG) fingerprint region. a** Experimental setup showing the LAO/STO sample (cooled to 8 K) and the metal-coated AFM tip oscillating at tapping frequency $\Omega_{tip}$, illuminated by a new fast-tunable narrow-band mid-infrared laser. **b** Real (solid lines) and imaginary part (dashed lines) of the dielectric functions of STO (blue) and LAO (red). Both oxides show phonon resonances in the mid-infrared range. **c** STO single crystals (left) were epitaxially covered with 8 uc of LAO. Depending on the termination of the STO substrate, the interface is either insulating (SrO-terminated) or conducting (TiO$_2$-terminated), the latter hosting a 2DEG. **d** Simulation of the Au-normalized near-field amplitude spectra $s_2/s_2^{Au}$ for STO substrate (blue), insulating LAO/STO (violet), and LAO/STO with 2DEG (dashed orange) at room temperature. Grey-shaded areas indicate spectral regions that have been investigated at room temperature before[22–24]. **e** Simulation of $s_2/s_2^{Au}$ at cryogenic temperatures (with increased 2DEG mobility). The shaded areas indicate the spectral range investigated at low temperatures previously[22] (grey) and in this work (green). **f** Zoom-in of Fig. 1d to low scattering amplitudes in the fingerprint region, showing the insulating (violet) and conducting (dashed orange) LAO/STO samples. **g** By referencing the conducting to the insulating case (dividing of scattering amplitude), changes introduced by the 2DEG are highlighted, resulting in a characteristic response with three separate maxima (fingerprint spectrum).

systems GmbH, with the samples being at a constant temperature of 8 K. For illumination, the narrow-band (linewidth <5 cm⁻¹) laser PT277-XIR by EKSPLA was used, which is based on a picosecond optical parametric oscillator with difference frequency generation and exhibits a high signal-to-noise ratio in the fingerprint region of the LAO/STO 2DEG. While the spot size of the focused laser on the sample is diffraction-limited to $\lambda/2$ (several micrometers in mid-IR), the near-fields at the tip apex depend on the tip radius and enable a lateral resolution down to the ten-nanometer range. The back-scattered light is demodulated at higher harmonics $n\Omega_{tip}$ of the tapping frequency, and pseudo-heterodyne interferometric detection allows for separation into near-field scattering amplitude $s_n$ and phase $\phi_n$. In s-SNOM, the absolute scattering signal strongly depends on the experimental setup, e.g., the tip shape, oscillation amplitude, detector sensitivity, or laser power. To acquire setup-independent results, it is an established technique to use normalization to a reference sample, resulting in the relative scattering amplitude $s_n/s_n^{ref}$ and phase $\phi_n-\phi_n^{ref}$.

This near-field scattering amplitude and phase depend on the frequency-dependent optical properties of the sample, expressed as the dielectric function or permittivity $\varepsilon(v)$. Figure 1b shows the real (solid lines) and imaginary (dashed lines) parts of the permittivity of bulk STO (blue) and bulk LAO (red), both of which exhibit phonon resonances in the mid-IR range. The near-fields of s-SNOM decay exponentially with distance to the tip, in the order of several ten nanometers[25–27], thus even layers of a few nanometers thickness can significantly influence the scattering signal[28–30], especially in the vicinity of zero-crossings of the materials' $Re[\varepsilon]$[23,31,32]. The probing depth also depends on the experimental parameters[33], and can thus be adjusted between several tens and hundreds nanometers. For our layered system, this leads to a combined scattering response of the STO substrate, the LAO top layer, and the 2DEG in between, dependent on their respective permittivities. To isolate the influence of the 2DEG layer from the contributions of the LAO covering layer and STO substrate, a direct comparison of samples with and without 2DEG, respectively, is helpful, ideally showing identical lattice phonon properties. This can be achieved by controlling the termination of the STO substrate (Fig. 1c): SrO-terminated STO surfaces lead to an insulating interface (without 2DEG) after LAO deposition, while TiO₂-terminated surfaces lead to a conductive interface (with 2DEG), if a critical thickness of at least 4 unit cells (uc) of LAO is deposited epitaxially[5,24]. The samples used in this work were fabricated by pulsed laser deposition (PLD) of 8 uc (3 nm) LAO on single crystal STO substrates. All substrates were wet-etched to get pure TiO₂ termination, and then for one of them, a single SrO layer was applied by PLD, to invert the termination (cf. Methods for details)[34]. These two types of LAO/STO samples fabricated in identical conditions from a split STO substrate enable a comparable measurement of conductive and insulating interface. This allows for a direct investigation of the 2DEG response at low temperatures by using the non-conducting interface as a reference in the s-SNOM measurement without introducing referencing artefacts.

To showcase the influence of the different layers, Fig. 1d presents simulations of the gold-normalized near-field scattering amplitude $s_2/s_2^{Au}$ for bulk STO (blue), LAO/STO without 2DEG (violet), and LAO/STO with 2DEG (dashed orange), at room temperature (cf. ref. 23). For all simulations and measurements, the demodulation order $n = 2$ is presented here, to get a high signal-to-noise ratio with sufficient background suppression in the observed spectral range. Simulations were done using the finite dipole model[35] for the near-field calculations, combined with the transfer matrix method to get the near-field response of an arbitrary layer stack (cf. Methods for details)[36]. The 2DEG was modeled as several layers of exponentially decaying carrier density[22], in accordance with depth profile assumptions from literature[37,38]. The STO substrate alone (blue) shows a strongly enhanced scattering amplitude ("near-field

resonance") slightly below its LO-frequency $v_{LO} = 788$ cm⁻¹, where $Re[\varepsilon_{STO}]$ is slightly negative and $Im[\varepsilon_{STO}]$ is low. When adding the ultra-thin LAO on top of STO (violet), the overall shape of the STO near-field resonance persists but is modified by the zero-crossings of $Re[\varepsilon_{LAO}]$ around 600, 650, and 750 cm⁻¹, resulting in amplitude features with derivative line shape. Adding 2DEG charge carriers at the interface (dashed orange), the scattering amplitude of the near-field resonance around 700 cm⁻¹ is reduced by additional damping and the high-frequency scattering above 900 cm⁻¹ is slightly increased (barely visible at this scale). Grey-shaded areas indicate spectral regions where the 2DEG was experimentally investigated in previous studies. These studies were limited to the influence of the 2DEG on i) the phonon near-field resonance of STO (<750 cm⁻¹), where the s-SNOM signal is generally high[23], or ii) the off-resonance scattering response in the spectral window of light sources with a high signal-to-noise ratio, such as CO₂ lasers (>920 cm⁻¹)[22,24]. In both regions, the general influence of free charge carriers on near-field spectra is only indirect, either i) as an additional source of damping (increased $Im[\varepsilon]$) or ii) as a constant background (increased high-frequency limit $\varepsilon_\infty$, cf. Supplementary Information S3). Thus, a clear separation of carrier concentration and mobility was not previously possible.

## Near-field fingerprint spectroscopy in LAO/STO

Figure 1e shows how the calculated near-field spectrum of the conducting LAO/STO interface (dashed orange) is expected to change at low temperatures, where the higher 2DEG mobility leads to a slight shift of the peak to higher frequencies and an increased scattering amplitude. Zooming-in to frequencies above the phonon near-field resonance, Fig. 1f shows significant differences between the scattering amplitude of the SrO-terminated (without 2DEG, violet) and the TiO₂-terminated LAO/STO interface (with 2DEG, dashed orange). Adding the 2DEG leads to three changes in the spectrum, as indicated by black arrows: (I) a shift of the high-frequency flank of the STO phonon near-field resonance to higher frequencies, (II) an additional spectral feature around 775 cm⁻¹, and (III) a general increase in scattering amplitude. However, the overall s-SNOM signal in the spectral range above 750 cm⁻¹ is quite low, with relative scattering amplitudes between 2% and 30% of that of gold (cf. Fig. 1d, e), and thus a high signal-to-noise ratio of the illumination source is necessary to record useable near-field spectra. This is especially true for our proposed normalization procedure between samples with and without 2DEG, respectively, which is the most sensitive to the 2DEG properties but also the most susceptible to artefacts from referencing to noise.

Figure 1g showcases the theoretical prediction of this characteristic 2DEG near-field response, which can be obtained by referencing the sample with 2DEG to the sample without 2DEG, i. e. by using Fig. 1e and dividing the data plotted in dashed orange by that in solid violet. The resulting normalized amplitude spectrum shows only changes to the spectrum induced by adding the 2DEG charge carriers to the non-conducting interface, with three distinct peaks that relate to the described changes I-III, introduced in Fig. 1e. As the height, position and shape of these features are very sensitive to changes of the electronic properties (cf. Fig. 2), they can be seen as a characteristic fingerprint near-field spectrum of this 2D electron system, similar to characteristic vibrational bands of polymers in near-field spectroscopy[39]. We have thus identified a unique fingerprint signature of the LAO/STO 2DEG, with three characteristic peaks in the spectral range of 700 to 900 cm⁻¹. Note that this fingerprint spectrum is constrained to the shown frequency range by the phonon properties of the two materials and could not be experimentally investigated in previous publications due to a lack of light sources with sufficient signal-to-noise ratio (cf. Fig. S1 in the Supplementary Information).

To illustrate how these fingerprint spectra of the 2DEG (cf. Fig. 1g) are influenced by the electronic properties, Fig. 2 shows theoretical

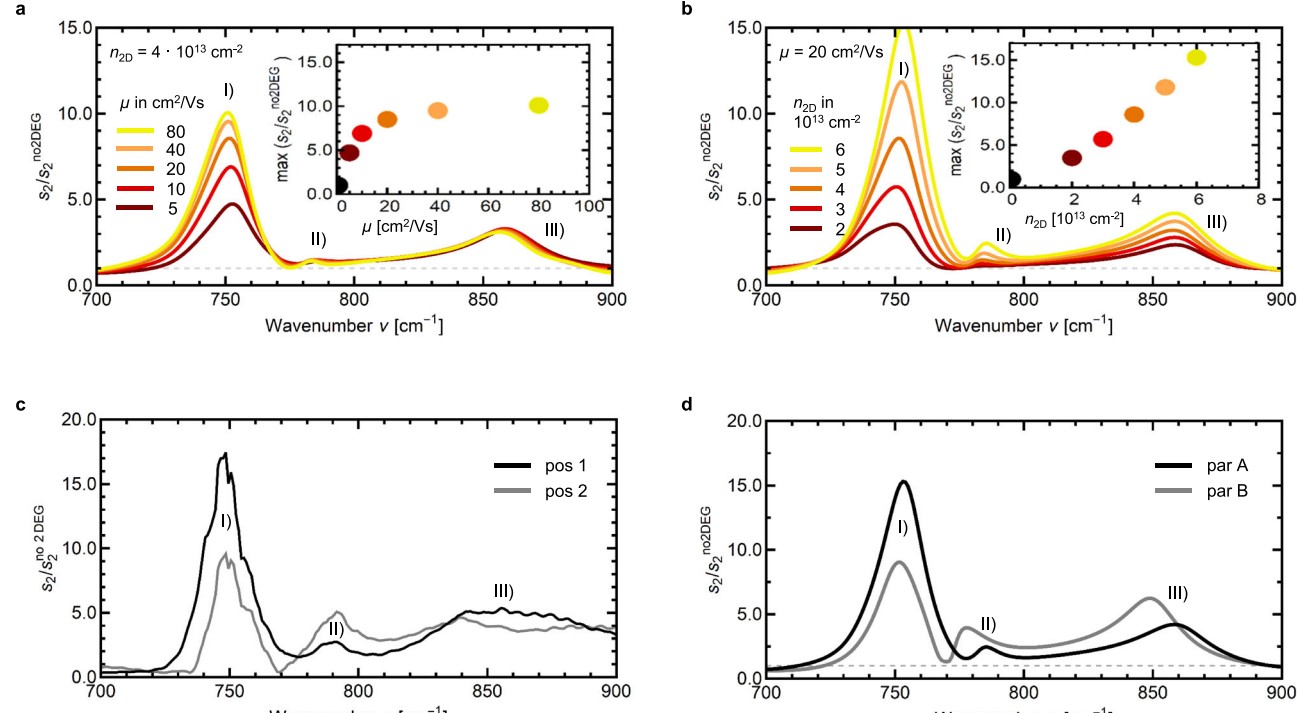

**Fig. 2 | Fingerprint spectra of the 2DEG at conducting LAO/STO interfaces, normalized to the non-conducting interface. a** Simulation for 2DEG mobilities $\mu$ between 5 cm²/Vs (dark red) and 80 cm²/Vs (yellow), with a constant 2DEG carrier concentration $n_{2D} = 4 \times 10^{13}$ cm⁻². The inset shows the maximum value of peak I) with rising mobility. **b** Simulation for different $n_{2D}$ between $2 \times 10^{13}$ cm⁻² (dark red) and $6 \times 10^{13}$ cm⁻² (yellow), with a constant $\mu = 20$ cm²/Vs. The inset shows the maximum value of peak I) with rising carrier concentration. **c** Measurement of fingerprint spectra at two different positions of the conducting sample. **d** Simulation of fingerprint spectra with different simulation parameter sets (cf. Supplementary Information S3), reproducing the experimental spectra shown in **c**.

predictions of the changes introduced by a varying 2DEG mobility $\mu$ (Fig. 2a) and 2DEG sheet carrier concentration $n_{2D}$ (Fig. 2b). Values for carrier concentration and mobility were picked in accordance with Hall measurements of the sample (cf. Supplementary Information S4), with experimental values of $n_{2D} = 4.3 \times 10^{13}$ cm⁻² and $\mu = 5.1$ cm²/Vs at room temperature. While the 2DEG mobility can increase to above $10^4$ cm²/Vs upon cooling[5], it should be noted that the effective mobility in the mid-infrared spectral range can be significantly lower than those measured in the far-infrared spectral range or in transport measurements, due to spectral weight redistribution from polaronic contributions[38,40]. In Fig. 2a, a rising mobility (transition from dark red to yellow) from 5 to 80 cm²/Vs leads to an increase of the relative scattering amplitude for the peak around 750 cm⁻¹, while the other two peaks remain mostly unchanged. Contrary to that, a rise in carrier concentration (transition from dark red to yellow in Fig. 2b) leads to an increase of the relative scattering amplitude for all three peaks. Additionally, the scaling behavior is very different, as can be seen from the insets in both plots, which show the maximum height of peak (I) with rising mobility or carrier concentration, respectively. Even though $\mu$ is doubled between each curve in Fig. 2a, the increase in peak height diminishes with each step, and saturation is reached around 80 cm²/Vs (yellow curve). In comparison, $n_{2D}$ is only increased linearly and the increase in peak height of the relative scattering signal persists across the whole value range shown here. Thus, the characteristic peaks of the near-field fingerprint spectrum are predicted to behave differently for the two parameters, finally allowing for direct experimental access to separated $n$ and $\mu$.

Figure 2c presents s-SNOM measurements of the 2DEG fingerprint region, for two different positions (black, grey) on LAO/STO with 2DEG (cf. Methods and Supplementary Information for details). For the first position (black curve), a high peak (I) around 750 cm⁻¹, a

small peak (II) at 790 cm⁻¹ and a broad peak (III) around 850 cm⁻¹ can be observed, which generally fits well to the theoretical predictions presented in Fig. 2a/b, especially for peak (I) and (II). The fingerprint spectrum recorded at the second position (grey) deviates significantly from the first, with a lower intensity in the first (I), a higher intensity in the second (II), and a less defined shape in the third peak (III). Additionally, the minimum around 775 cm⁻¹ and the maximum around 850 cm⁻¹ are both shifted to lower frequencies by 5–10 cm⁻¹. Figure 2d presents simulations with different parameter sets, to reproduce the measurements from Fig. 2c: set A (black) was achieved with $n_{2D} = 6 \times 10^{13}$ cm⁻² and $\mu = 20$ cm²/Vs, while set B (grey) relates to $n_{2D} = 8 \times 10^{13}$ cm⁻² and $\mu = 10$ cm²/Vs. Generally, good agreement between 2DEG fingerprint spectra and simulations can be achieved, showing that the theoretical description of the 2DEG with a finite dipole model, transfer matrix method, exponentially decaying depth profile, and phonon background represents the physical behavior of the system well. The first parameter set (black) reproduces the first measurement position (black in Fig. 2c) well for the first (I) and second (II) peak. Generally, peak (III) seems much broader in the measurement, which might result from frequency-dependent damping $\gamma(v)$ for the free charge carriers around the LO-frequency of the STO substrate[41], which is not yet well understood and not included in the model. The higher background at 900 cm⁻¹ in both measurements compared to the respective simulations could be explained by a change in the high-frequency limit $\varepsilon_\infty$ of the dielectric function. Such a change in $\varepsilon_\infty$ was shown to influence STO near-field signals upon doping[42,43], possibly also contributing to the broader peak (III).

For the second parameter set (grey), the goal was to reproduce the main characteristics of this near-field fingerprint spectrum, i.e., the different intensity distribution between the first two peaks (I) and (II),

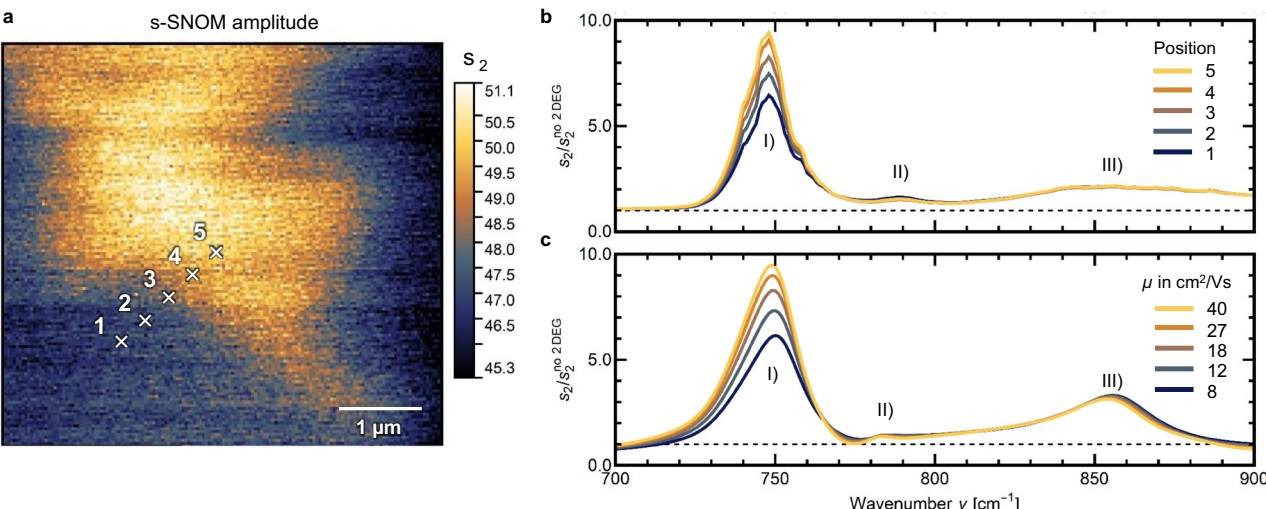

**Fig. 3 | Fingerprint investigation of lateral inhomogeneities on the nanoscale.** **a** s-SNOM image of near-field amplitude $s_2$ at $v = 700\ cm^{-1}$, close to the maximum of the scattering response from the STO phonon near-field resonance (cf. Fig. 1b–d). Markers 1–5 indicate measurement positions of a line scan with the resulting spectra shown in Fig. 3b. **b** Fingerprint spectra measured at positions 1 (blue) to 5 (yellow) of Fig. 3a. **c** Simulation of fingerprint spectra with 2DEG mobility µ rising from 8 cm²/Vs (blue) to 40 cm²/Vs (yellow).

as well as the shift of the minimum between these two peaks to lower frequencies. However, this matching was not possible by changing the electronic properties alone. Instead, small changes to the LAO/STO phonon properties were necessary to achieve the frequency shift of the minimum (cf. Supplementary Information S3) and the third maximum (III). This points towards inhomogeneities of the LAO layer, possibly due to differently strained areas after cooling, as strain can lead to a significant shift of the phonon frequencies in oxides[44]. The inhomogeneous 2DEG properties also detected by this method could result from magnetic or structural domains observed for LAO/STO 2DEGs at low temperature[45–48].

To further investigate how these near-field fingerprint spectra (cf. Fig. 2c) vary for different positions on the sample, we performed preliminary imaging (Fig. 3a) at a frequency of 700 cm⁻¹, where the scattering signal is highest due to the strong phonon near-field response (cf. Fig. 1d, e). These single-frequency s-SNOM images show significant variations in amplitude $s_2$ (Fig. 3a) and phase $\phi_2$ (Supplementary Information S5). This local inhomogeneity was further investigated with near-field spectra at measurement positions 1–5 indicated by crosses in Fig. 3a. The recorded spectra were again normalized to the insulating interface and the results are presented in Fig. 3b, showing the previously discussed fingerprint spectrum of the 2DEG, with a high (I), a small (II), and a broad peak (III). Interestingly, moving from measurement position 1 (dark blue) to 5 (yellow), the intensity of the first peak (I) increases by 50%, while the other two peaks remain mostly unchanged. This behavior is very similar to Fig. 2a and can be reproduced by a variation of the 2DEG mobility µ between 8 cm²/Vs and 40 cm²/Vs (Fig. 3c), showing that with near-field fingerprint spectroscopy, the local inhomogeneities observed in the s-SNOM image (Fig. 3a) can be linked to a laterally varying 2DEG mobility. Additional s-SNOM images recorded after the line scan (cf. Supplementary Information S5) indicate that the inhomogeneous mobility could be affected by the previous measurement of the same area. An influence of low-temperature scanning probe measurements on the electron mobility of LAO/STO is known from literature and a current topic of research, e.g., arising due to electrostatic gating from the irradiated AFM tip[22], persistent photoconductivity[49,50], or frozen condensates[51]. Using the increased sensitivity of near-field fingerprint spectroscopy to the local electronic properties, the physical origin of such variations can now be investigated in the future.

## Generalized near-field fingerprint spectroscopy

We have shown that near-field fingerprint spectroscopy is a method to investigate the electronic properties of the 2DEG in LAO/STO heterostructures with high sensitivity, allowing to finally separate influences of $n$ and µ in s-SNOM. Now, we explore the underlying reasons for the high sensitivity of near-field fingerprint spectroscopy to the 2DEG by evaluating the working principle of s-SNOM step-by-step. In s-SNOM, the frequency range around zero crossings of Re[$\varepsilon$] has the highest sensitivity $\delta s_n/\delta \varepsilon$ to small changes of the optical properties. Figure 4a shows the calculated s-SNOM coupling function, i.e., the scattering response of the tip-sample coupled system in dependence of the (bulk) sample dielectric function $\varepsilon$. Here, the scattering amplitude normalized to a constant Au-reference $s_2/s_2^{Au}$ is plotted against Re[$\varepsilon$] in the vicinity of its zero-crossing, with different curves representing different values of the imaginary part, from Im[$\varepsilon$] = 0.5 (light grey) to Im[$\varepsilon$] = 16 (black). The coupling function shows a strongly non-linear behavior between Re[$\varepsilon$] = −10 and +10, with a maximum around −4 and a minimum around +1. This behavior is typical for dipolar near-field coupling and also applicable to the finite dipole model used here. The red-shaded area highlights the range of highest sensitivity $\delta s_n/\delta \varepsilon$, with a zoom-in shown Fig. 4b. The green arrow indicates a shift to lower Re[$\varepsilon$], that would be expected from additional free charge carriers (cf. Supplementary Information S3), such as the 2DEG in LAO/STO.

Beyond LAO/STO, near-field fingerprint spectroscopy is sensitive to different physical effects, such as the LO-frequency of phonons, the plasma frequency of free charge carriers, or band gaps. This generalizability is presented in Fig. 4c–f using the example of doped InAs, which has a doping-dependent zero-crossing of Re[$\varepsilon$] at its "screened" plasma frequency, i.e., shifted compared to the free electron plasma due to the semiconductor background. Doped InAs also exhibits a surface accumulation layer, which confines a higher concentration of free charge carriers close to the interface to air[52]. Fig. 4c/d present dielectric functions of a heavily doped (bulk) InAs sample (dark green, $n_{bulk} = 1.6 \times 10^{19}\ cm^{-3}$) as well as its surface accumulation layer (light green, $n = 2.0 \times 10^{19}\ cm^{-3}$)[53]. The higher carrier concentration leads to an increase of the screened plasma frequency by $\Delta v_p \approx 22\ cm^{-1}$, as well as lower Re[$\varepsilon$] and higher Im[$\varepsilon$] at any frequency.

The simulated scattering amplitude $s_2/s_2^{Au}$ of a homogeneously doped bulk InAs sample is shown in Fig. 4e (dark green). Adding the ultrathin ($d = 3.6\ nm$) surface accumulation layer on top of the bulk

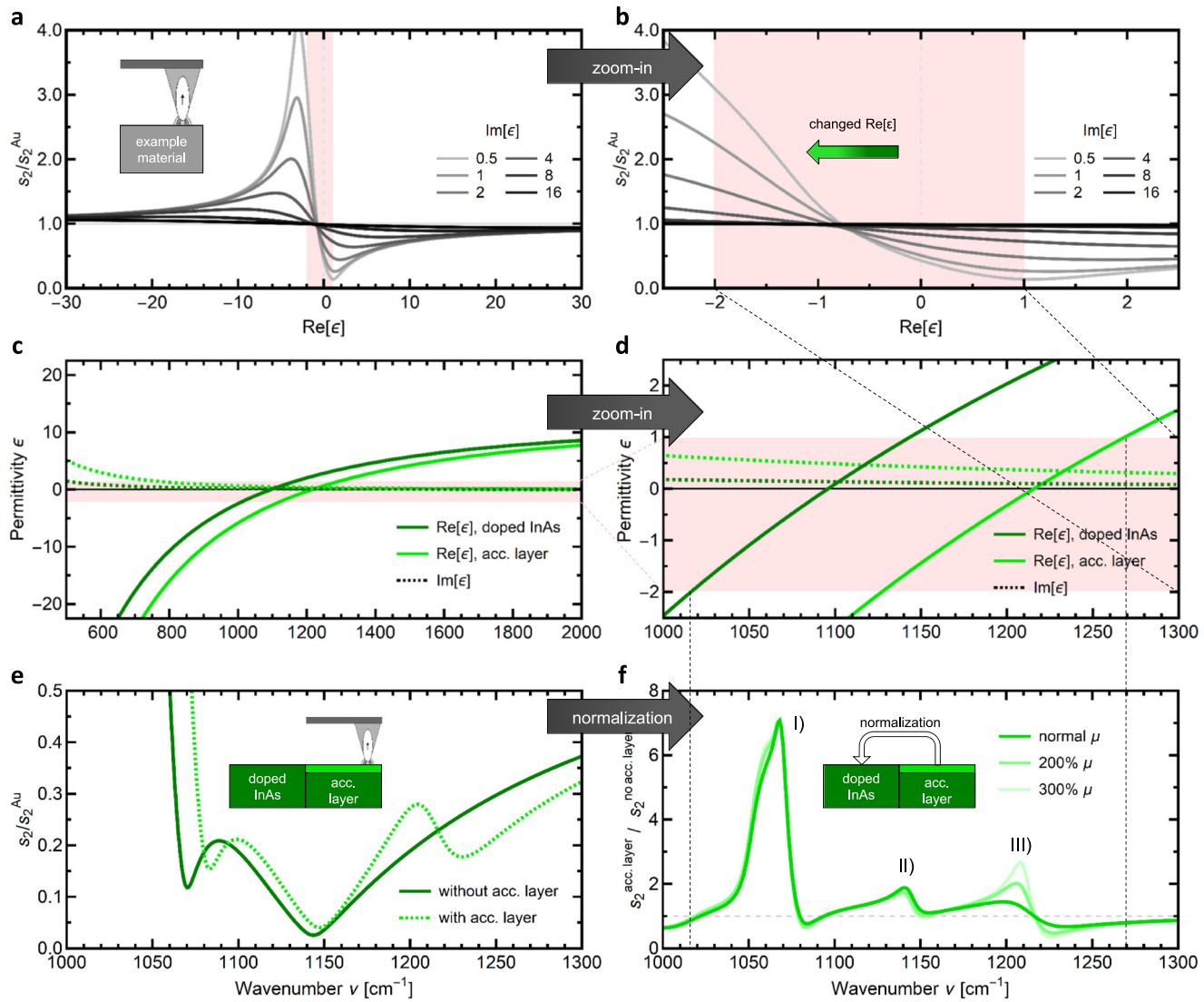

**Fig. 4 | Generalization of near-field fingerprint spectroscopy, using the example of surface accumulation layers in doped InAs. a** Calculated coupling function $s_2(\varepsilon)$ of s-SNOM for model materials with different imaginary part (light grey to black for rising $Im[\varepsilon]$), the red shaded area highlights the range of highest slope. **b** Zoom-in highlighting the high sensitivity $\delta s_n/\delta \varepsilon$ in this range (cf. Supplementary Information S2), which is the origin of near-field fingerprint spectra. **c** Dielectric function $\varepsilon(v)$ (cf. Supplementary Information S3) of bulk doped InAs (dark green, $n = 1.6 \times 10^{19}$ cm$^{-3}$) and InAs surface accumulation layer (light green, $n = 2.0 \times$ $10^{19}$ cm$^{-3}$). The red shaded area highlights the range of highest s-SNOM sensitivity $\delta s_n/\delta \varepsilon$. **d** Zoom-in showing the shift of real and imaginary part. **e** Near-field amplitude spectra $s_2/s_2^{Au}$ for bulk doped InAs (dark green) and an InAs sample with surface accumulation layer of higher carrier concentration (dashed light green). **f** Normalized near-field fingerprint spectra showcasing the changes introduced by the surface accumulation layer for different carrier mobilities μ (lighter green). The peaks (I), (II), and (III) are labeled similarly to the 2DEG in Figs. 1–3.

(dashed light green curve), leads to a small shift of the flank and modifications to the shape, similar to the influence of the 2DEG in LAO/STO (cf. Fig. 1f). Therefore, if a sample with accumulation layer is normalized to a sample without accumulation layer (Fig. 4f), the near-field fingerprint spectrum shows the characteristic differences introduced by the ultrathin surface modification. The InAs fingerprint spectrum is very similar to those shown for LAO/STO, with a main peak I) and two smaller peaks II) and III). However, increasing the electron mobility μ (towards lighter green) mainly results in changes to peak III) for InAs, whereas it mainly influenced peak I) for the LAO/STO 2DEG (cf. Fig. 3), showing that near-field fingerprint spectra are characteristic for the specific material and layer geometry. This can be used for a generalization to many different materials, even in bulk materials without surface layers (cf. Supplementary Information S2). Self-normalization between different states of the material leads to an increased number of characteristic points

(minima, maxima, and zero crossings) that improve the quality of fitting procedures. For this, it is important to note that a description of sample materials and geometry must be built into the theoretical model to understand the link between s-SNOM signal and material properties. As such, additional knowledge from correlative measurements is necessary to gain meaningful insight into the near-field fingerprint spectrum. The resulting pattern, however, is highly sensitive to small changes in optical properties resulting from local inhomogeneities, surface modifications, or deliberately influencing the material (e.g., via gating), making s-SNOM with near-field fingerprint spectroscopy a powerful tool for nanoscale material analysis.

## Discussion
We presented a method to use s-SNOM with novel light sources to investigate nanoscale inhomogeneities in materials for modern

electronic applications, by using near-field fingerprint spectra resulting from self-normalization. These near-field fingerprint spectra are highly sensitive to changes of the electronic properties, due to high sensitivity $\delta s_n/\delta\varepsilon$ to the dielectric function in the fingerprint spectral range. We used this method to theoretically predict the ideal spectral region for the s-SNOM investigation of the 2DEG in LAO/STO at low temperatures. Three characteristic peaks can be found, which are differently influenced by 2DEG carrier concentration and mobility. The changes introduced by 2DEG mobility mostly relate to the main fingerprint peak around 750 cm⁻¹, with a saturation of the influence around μ = 80 cm²/Vs. The influence of carrier density is stronger by comparison and influences all three peaks similarly. This now allows for the separation of carrier concentration and carrier mobility, which can influence s-SNOM spectra in similar ways. Using a newly-developed light source in combination with cryo-s-SNOM at 8 K, we were then able to measure near-field fingerprint spectra of the 2DEG by comparing samples of different substrate terminations in a previously inaccessible spectral window with pseudo-heterodyne point spectroscopy. We found local variations in measured fingerprint spectra, which could be explained by changes of the 2DEG and phonon properties. We also presented the general applicability of our method by predicting the mobility-dependent fingerprint response of surface accumulation layers in doped InAs.

In the future, this method can be used to disentangle the local carrier concentration and mobility in investigations of the 2DEG in LAO/STO, for example, for the characterization of different types of defects[54], such as varying LAO thickness, atomic steps, or vacancies of different ions. Additionally, the higher 2DEG mobility at low temperature increases the surface sensitivity of s-SNOM compared to room temperature and should allow for conclusions about the vertical distribution[38,55] of the charge carriers. More generally, novel light sources allow fast frequency sweeping across a broad spectral range in the infrared region with a high signal-to-noise ratio. This enables self-normalization even at low scattering amplitudes, as was successfully shown for interband transitions in tetralayer graphene recently[56]. Upon identification of promising characteristic frequencies, s-SNOM images can be recorded using a narrow linewidth[42], to investigate local defects with nanoscale resolution at the frequency where the contrast is strongest. Here, we have demonstrated a link between quantitative analysis and spatially resolving carrier density and mobility in the 2DEG of LAO/STO via near-field fingerprint spectra. This method can be easily transferred to other confined correlated electron systems such as 2DEGs at different (oxide) heterointerfaces[57,58], van-der-Waals heterostructures or multilayers of 2D materials[56,59], or topological insulators[60,61], providing spatially resolved information for designing nanoelectronic devices.

## Methods

### Sample preparation

Eight unit cells of LaAlO₃ were deposited on wet-etched, TiO₂-terminated (100)-SrTiO₃ substrates using pulsed laser deposition (PLD). The films were deposited at a laser frequency of 1 Hz at a fluence of 1.0 J/cm² and an oxygen pressure of 1 × 10⁻⁴ mbar. The growth temperature was 800 °C and the samples were quenched down to room temperature after a relaxation at growth temperature[62]. During growth, clear RHEED-intensity oscillations were observed, indicating a layer-by-layer growth mode and yielding single unit cell thickness control. The samples were characterized by AC-Hall effect measurements in van-der-Pauw geometry at room temperature indicating a sheet carrier density of 4.3 × 10¹³ cm⁻² and an electron mobility of 5.1 cm²/Vs. SrO termination was achieved by PLD deposition in the same system as for the LAO thin films from a ceramic SrO₂ target using a laser fluence of 0.9 J/cm². The chamber was held at an oxygen partial pressure 2 × 10⁻⁷ mbar and the substrate was heated to 800 °C. To achieve the change of termination, the deposition was stopped as soon

as the first order diffracted spot fulfilled one oscillation, crossing the intensity of the specular spot[63].

### s-SNOM measurements

s-SNOM measurements were done using a commercial low-temperature scattering-type scanning near-field optical microscope (cryo-neaSCOPE by attocube systems GmbH), with a pseudo-heterodyne detection configuration[64] for single-frequency 2D imaging and sequential point spectroscopy. As a light source, a commercial narrow-band tunable laser (EKSPLA PT277-XIR) was used, consisting of a tunable optical parametric oscillator (OPO) combined with difference frequency generation (DFG), that delivers beams of approx. 10 mW power in the DFG region (625–2000 cm⁻¹), in 8 ps pulses with 87 MHz repetition rate and a typical line width of <3 cm⁻¹ in this frequency range. As a detector, a photoconductive Mercury Cadmium Telluride (HgCdTe) detector element of 50 μm diameter with a ZnSe window was used. As scattering tips, commercial nano-FTIR probes by attocube systems GmbH were used, with a tapping amplitude of approximately 80 nm. For in-situ referencing, both samples (with and without 2DEG, respectively) were glued side by side on a copper plate mounted on the sample stage. The system was cooled down to its base temperature with the samples being stabilized at $T = 8$ K, as monitored by a calibrated Cernox sensor, which was thermally coupled to the sample plate. Cooling was provided by an integrated closed-cycle cryostat (attoDry800 from attocube systems GmbH), keeping the sample space free of cryogenic media.

### Theoretical modeling

The Finite Dipole Model (FDM)[35] and Transfer Matrix Method (TMM) were used in combination[36] to determine the near-field scattering response of arbitrarily layered systems with known dielectric functions. FDM parameters were 400 nm ellipsoid length, 90 nm tip radius, 80 nm tapping amplitude, demodulation order $n = 2$, and the geometric factor $g = 0.7 \times \exp(0.1i)$. The p-polarized TMM reflection coefficient was used as the FDM sample reflection factor[36], at a dominant in-plane wavevector[29] of $\mathbf{k}_x = 250\,000$ cm⁻¹. To model the optical properties of the layer stack, the dielectric functions of STO and LAO were calculated from literature data[65,66] using the Berreman-Unterwald-Lowndes factorized form[41,67] of multiple Lorentz oscillators (cf. Supplementary Information S3). To describe the dielectric function of the 2DEG, a Drude term was added to the STO phonon background[41], which was then used in a multilayer approach of 10 slices with thickness $d = 1$ nm and exponentially decaying charge carrier concentration away from the interface, distributing the 2D sheet carrier density with a decay constant $z_0 = 2$ nm. The effective mass was approximated to an averaged effective mass of $m^* = 3.2\ m_0$ (with the free electron mass $m_0$) as predicted by First Principles calculations[55] and measured experimentally[38].

## Data availability

Data sets generated during the current study are available from the corresponding author on request.

## Code availability

Code used for theoretical calculations in the current study is available from the corresponding author on request.

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

## Acknowledgements
J.B., K.W., and T.T. acknowledge funding from the Deutsche Forschungsgesellschaft (DFG) within the collaborative research center SFB 917 and under grant agreement no. TA 848/7-1. M.A.R. and F.G. acknowledge financial support from the Deutsche Forschungsgesellschaft (DFG) FG 1604 (No. 315025796).

## Author contributions
J.B., T.T., and F.G. conceived and planned the study. J.B., Y.C.D., and R.H. planned and prepared the measurements, which J.B., T.T., Y.C.D., and R.H. carried out. M.A.R. and F.G. provided the samples; J.B., M.A.R., and F.G. did pre- and post-characterization. J.B. and K.W. performed data evaluation and simulations. All authors discussed the data and contributed to writing the paper. All authors approved the final version of the manuscript.

## Funding

## Competing interests
Y.C.D. and R.H. are employees of attocube systems GmbH, producer of the cryo-s-SNOM microscope used in this study. The remaining authors declare no competing interests.
