## [Transparent Peer Review file · Nature Communications]

Low temperature near-field fingerprint spectroscopy of 2D electron systems in oxide heterostructures and beyond

Corresponding Author: Mr Julian Barnett

Version 0:

Reviewer comments:

Reviewer #1

(Remarks to the Author)

This paper addresses near-field spectroscopy of the 2D electron gas at the interface of LAO/STO. With careful modeling and selection of probe wavelength, the authors demonstrate that they can separate local charge carrier density and mobility and monitor their variations across the sample. These are very challenging experiments, requiring spectral near-field measurements under low temperature conditions, and the modeling is carefully performed. The spatial variation resolved experimentally is interesting and demonstrates the sensitivity of the tool. However, the current form of the paper seems to be of primarily niche interest, relating to demonstration of the analysis on a “model” LAO/STO system. While the approach may be more generalizable, there is not much discussion of this, such as the conditions under which it is possible, and the accuracy of the resulting properties extracted. The paper would be strengthened by some corroboration of the mobilities and carrier concentrations extracted through another technique, or by studying several samples prepared under variable conditions. For example, in the conclusions, the authors suggest “varying LAO thickness, atomic steps, or vacancies of different ions”.

Parts of the description of the “fingerprint” region for particular layers would also benefit from further clarification. For example, the authors suggest that s-SNOM possesses both high surface sensitivity and sub-surface capabilities, which appears contradictory, and would benefit from greater discussion of when either regime would be probed.

Reviewer #2

(Remarks to the Author)

Manuscript entitled “Mid-infrared near-field fingerprint spectroscopy of the 2D electron gas in LaAlO₃/SrTiO₃ at low temperatures” by Julian Barnett et. al., presents a detailed study on exploiting the theoretical predictions of a characteristic spectral region for the s-SNOM investigation of the LAO/STO 2DEG at low temperatures. In summary, the experimental data are of high quality and well backed-up by supporting information. I find this paper to be very-well conceived and nicely written. I recommend publication in Nature Communication after some changes.

1. In Figures 2a and b, the authors have shown the variation in the intensity of peaks depending on the mobility and carrier density. But the physical explanations is lacking in the manuscript. Also the reasons why the mobility change only in the first peak(I), but the carrier density change in all three peaks (I,II,III). It would be important to include an hypothesis (possible origins) in a qualitative way.
2. While the variation of carrier density and mobility could result in clear variation based on simulation (Fig. 2a and b), it's difficult to link the spectra to the exact carrier density and mobility for an unknown system (LAO/STO is a known-answer question), especially when there are extrinsic contributions in play, such as defects, local inhomogeneous, etc. Therefore, one would have a feeling that other approaches (e.g., Hall measurement) are more reliable.
3. The results in Fig. 2c and d are not convincing. The mechanism for the shift and variation of peak II and III are not clear and the simulations probably should be optimized.
4. It's not mandatory, but it's suggested to remove Fig. s1 a & b, and put Fig. s1c and d as insets in Fig. 2 a and b, respectively.
5. As observed by the authors, ‘the measurement itself could influence the local electronic properties (line 233, Fig. s2)’. One relevant question is, how long it takes for the sample to fully recover to a ‘fresh’ state? If this is not clear, one would expect that all the spectra recorded are not for ‘fresh’ state (and the intrinsic cannot be reached by this method)? Above which

sweeping frequency could the measurements be reliable?

Reviewer #3

(Remarks to the Author)

In the present manuscript, the authors show mid-infrared near-field fingerprint spectroscopy of the two-dimensional electron gas (2DES) at the LaAlO₃/SrTiO₃ (LAO/STO) interface at 8 K, and further obtain the carrier density and mobility by theoretical fitting. The results are interesting, but do not provide any new insight into the nature of the 2DES at the LAO/STO interface. Actually, the LAO/STO 2DES have been extensively studied, especially by electronic transport, to quantitatively reveal its nature. Characterizations using near-field spectroscopy similar to the present work have been performed for 2DES at oxide interfaces (include LAO/STO), see Nat. Commun. 10, 2774 (2019); Adv. Funct. Mater. 30, 2004767 (2020); Nano Lett. 22, 2755 (2022). These studies have also obtained successfully the carrier density and mobility, though the adopted wavelength range may be different from the present study. I fail to see any brand-new insight from the present study. The authors also demonstrate the lateral inhomogeneity by near-field imaging, which has been well established previously. And the near-field results in this study do not help to reveal the origin of the inhomogeneity, at least the authors do not discuss about. I therefore do not recommend publication of the present manuscript in Nature Communications.

More comments are listed below.

1. the hardcore experimental data are very limited, and the reproducibility is poor. For example, the two images scanned at the same area dramatically different, as shown in Fig. S2. The authors attribute such nonreproducibility to the changes in the local potential, which is not very convincing to me.
2. The authors adopt 700 cm⁻¹ for imaging (Fig. 3a), which is not any of the identified fingerprint peaks (750 cm⁻¹, 775 cm⁻¹, 860 cm⁻¹). Why? From Fig. 2, the intensity at 700 cm⁻¹ is not sensitive to the inhomogeneity at all.
3. It is better to compare the experimental data (Fig. 2c) and fitting data (Fig. 2d) directly in the same figure. It seems to me the fitting is not really good, considering it is a multi-parameter fitting.

Version 1:

Reviewer comments:

Reviewer #2

(Remarks to the Author)

As mentioned in my previous report, the manuscript titled 'Mid-infrared Near-field Fingerprint Spectroscopy of the 2D Electron Gas in LaAlO₃/SrTiO₃ at Low Temperatures' is a comprehensive and well-executed study. The authors have addressed all of my questions satisfactorily, and I have no further concerns. I recommend that the paper be published as it stands.

Reviewer #3

(Remarks to the Author)

The authors have made a great effort to address the comments from me and other referees. Their responses and revisions are satisfactory, and I recommend publication in Nature Communications.

Response Letter

Dear Reviewers,

we thank you for pointing out the raised issues, leading us to rework and improve the manuscript in significant ways. While focusing on different points, the reviewers fundamentally agreed that a more detailed discussion of our approach and its applicability to different material classes would be desirable, motivating us to strongly expand our explanation of near-field fingerprint spectroscopy. For the reviewer's convenience, we list and address all remarks point-by-point below and reproduce changes made to the manuscript in this letter. In the manuscripts (*Main Text* and *Supporting Information*), all changes are highlighted accordingly in blue. Changed Figures are marked with a blue frame. The reviewer comments are reproduced in orange.

Reviewer 1:

This paper addresses near-field spectroscopy of the 2D electron gas at the interface of LAO/STO. With careful modeling and selection of probe wavelength, the authors demonstrate that they can separate local charge carrier density and mobility and monitor their variations across the sample. These are very challenging experiments, requiring spectral near-field measurements under low temperature conditions, and the modeling is carefully performed. The spatial variation resolved experimentally is interesting and demonstrates the sensitivity of the tool.

1. **Point raised:** However, the current form of the paper seems to be of primarily niche interest, relating to demonstration of the analysis on a "model" LAO/STO system. While the approach may be more generalizable, there is not much discussion of this, such as the conditions under which it is possible, and the accuracy of the resulting properties extracted.

Answer: We thank the reviewer for acknowledging the challenging difficulty of the presented experiments and agree that we did not present the generalizability of our approach in a convincing manner. The principle of near-field fingerprint spectroscopy presented in our paper is applicable to different physical phenomena that lead to a zero crossings in $\text{Re}[\epsilon]$, such as plasmons, phonons, or interband transitions. Additionally, its high surface sensitivity makes it a potentially very useful method for material classes such as 2D materials or topological insulators, where interesting electronic properties arise from ultra-thin surface(-near) layers. We expanded the content of our manuscript to include simulations of an accumulation layer in doped InAs, which showcases the general applicability of the principle. To represent this in the manuscript, we included a completely new figure and expanded the main text significantly, changed the title to represent the broader scope and adjusted the abstract accordingly.

Action taken:

- A new figure (Figure 4, including figure caption) was added to the main text:

Figure 4: Generalization of near-field fingerprint spectroscopy, using the example of surface accumulation layers in doped InAs. **a** Calculated coupling function $s_2(\epsilon)$ of s-SNOM for model materials with different imaginary part (light grey to black for rising $\text{Im}[\epsilon]$), the red shaded area highlights the range of highest slope. **b** Zoom-in highlighting the high sensitivity $\delta s_n / \delta \epsilon$ in this range (cf. Supplementary Information S2), which is the origin of near-field fingerprint spectra. **c** Dielectric function $\epsilon(\nu)$ (cf. Supplementary Information S3) of bulk doped InAs (dark green, $n = 1.6 \times 10^{19} \text{ cm}^{-3}$) and InAs surface accumulation layer (light green, $n = 2.0 \times 10^{19} \text{ cm}^{-3}$). The red shaded area highlights the range of highest s-SNOM sensitivity $\delta s_n / \delta \epsilon$. **d** Zoom-in showing the shift of real and imaginary part. **e** Near-field amplitude spectra s_2/s_2^{Au} for bulk doped InAs (dark green) and an InAs sample with surface accumulation layer of higher carrier concentration (dashed light green). **f** Normalized near-field fingerprint spectra showcasing the changes introduced by the surface accumulation layer for different carrier mobilities μ (lighter green). The peaks I), II), and III) are labelled similarly to the 2DEG in Figures 1-3.

- Additional paragraphs were added to the main text, that describe the new Figure 4 and expand on the generalizability of the principle of near-field fingerprint spectroscopy:

We have shown that near-field fingerprint spectroscopy is a method to investigate the electronic properties of the 2DEG in LAO/STO heterostructures with high sensitivity, allowing to finally separate influences of n and μ in s-SNOM. Now, we explore the underlying reasons for the high sensitivity of near-field fingerprint spectroscopy to the 2DEG by evaluating the working principle of s-SNOM step-by-step. In s-SNOM, the frequency range around zero crossings of $\text{Re}[\epsilon]$ has the highest sensitivity $\delta s_n / \delta \epsilon$ to small changes of the optical properties. Figure 4a shows the generalized calculated s-SNOM coupling function, i. e. the scattering response of the tip-sample coupled system in dependence of an arbitrary (bulk) sample dielectric function ϵ . Here, the scattering amplitude normalized to a constant Au-reference s_2/s_2^{Au} is plotted against $\text{Re}[\epsilon]$ in the vicinity of its zero-crossing, with different curves

representing different values of the imaginary part, from $\text{Im}[\varepsilon] = 0.5$ (light grey) to $\text{Im}[\varepsilon] = 16$ (black). The coupling function shows a strongly non-linear behavior between $\text{Re}[\varepsilon] = -10$ and $+10$, with a maximum around -4 and a minimum around $+1$. This behavior is typical for dipolar near-field coupling and also applicable to the finite dipole model used here. The red-shaded area highlights the range of highest sensitivity $\delta s_n / \delta \varepsilon$, with a zoom-in shown Figure 4b. The green arrow indicates a shift to lower $\text{Re}[\varepsilon]$, that would be expected from additional free charge carriers (cf. Supplementary Information S3), such as the 2DEG in LAO/STO.

Beyond LAO/STO, near-field fingerprint spectroscopy is sensitive to different physical effects, such as the LO-frequency of phonons, the plasma frequency of free charge carriers, or band gaps. This generalizability is presented in Figure 4c-f using the example of doped InAs, which has a doping-dependent zero-crossing of $\text{Re}[\varepsilon]$ at its “screened” plasma frequency, i.e. shifted compared to the free electron plasma due to the semiconductor background. Doped InAs also exhibits a surface accumulation layer, which confines a higher concentration of free charge carriers close to the interface to air.⁵² Figure 4c/d present dielectric functions of a heavily doped (bulk) InAs sample (dark green, $n_{\text{bulk}} = 1.6 \times 10^{19} \text{ cm}^{-3}$) as well as its surface accumulation layer (light green, $n = 2.0 \times 10^{19} \text{ cm}^{-3}$).⁵³ The higher carrier concentration leads to an increase of the screened plasma frequency by $\Delta \nu_p \approx 220 \text{ cm}^{-1}$, as well as lower $\text{Re}[\varepsilon]$ and higher $\text{Im}[\varepsilon]$ at any frequency.

The simulated scattering amplitude s_2/s_2^{Au} of a homogeneously doped bulk InAs sample is shown in Figure 4e (dark green). Adding the ultrathin ($d = 3.6 \text{ nm}$) surface accumulation layer on top of the bulk (dashed light green curve), leads to a small shift of the flank and modifications to the shape, similar to the influence of the 2DEG in LAO/STO (cf. Figure 1f). Therefore, if a sample with accumulation layer is normalized to a sample without accumulation layer (Figure 4f), the near-field fingerprint spectrum shows the characteristic differences introduced by the ultrathin surface modification. The InAs fingerprint spectrum is very similar to those shown for LAO/STO, with a main peak I) and two smaller peaks II) and III). However, increasing the electron mobility μ (towards lighter green) mainly results in changes to peak III) for InAs, whereas it mainly influenced peak I) for the LAO/STO 2DEG (cf. Figure 3), showing that near-field fingerprint spectra are characteristic for the specific material and layer geometry. This can be used for a generalization to many different materials, even in bulk materials without surface layers (cf. Supplementary Information S2). Self-normalization between different states of the material leads to an increased number of characteristic points (minima, maxima, and zero crossings) that improve the quality of fitting procedures. For this, it is important to note that a description of sample materials and geometry must be built into the theoretical model to understand the link between s-SNOM signal and material properties. As such, additional knowledge (e.g. layer thickness or bulk dielectric function) from correlative measurements is necessary to gain meaningful insight into the near-field fingerprint spectrum. The resulting pattern, however, is highly sensitive to small changes in optical properties resulting from local inhomogeneities, surface modifications, or deliberately influencing the material (e. g. via gating), making s-SNOM with near-field fingerprint spectroscopy a powerful tool for nanoscale material analysis.

- We summarized our findings in a more general way in the conclusion:

We presented a method to use s-SNOM with novel light sources to investigate nanoscale inhomogeneities in materials for modern electronic applications, by using near-field fingerprint spectra resulting from self-normalization. These near-field fingerprint spectra are highly sensitive to changes of the electronic properties, due to high sensitivity $\delta s_n / \delta \varepsilon$ to the dielectric function in the fingerprint spectral range. We used this method to theoretically predict the ideal spectral region for the s-SNOM investigation of the 2DEG in LAO/STO at low temperatures. In this region (...). We also presented the

general applicability of our method by predicting the mobility-dependent fingerprint response of surface accumulation layers in doped InAs.

- Consequently, we also modified the title of our publication to decrease the focus on LAO/STO and better match our broader claim. The new title reads: “Low temperature near-field fingerprint spectroscopy of 2D electron systems in oxide heterostructures and beyond”
- Finally, we changed the abstract to better reflect this claim: “Finally, we model the surface accumulation layer in doped InAs, to show that our fingerprint spectra are a universal feature and generally applicable to confined electron systems, like topological insulators or stacked van-der-Waals materials.”

2. **Point raised:** The paper would be strengthened by some corroboration of the mobilities and carrier concentrations extracted through another technique, or by studying several samples prepared under variable conditions. For example, in the conclusions, the authors suggest “varying LAO thickness, atomic steps, or vacancies of different ions”.

Answer: We agree that corroboration with other methods would be desirable. We address this by now including additional Hall measurements of the sample in the Supplementary Information. However, due to limited access and the challenging conditions acknowledged by the reviewer in the previous point, further cryo-SNOM measurements are currently not possible for us.

Action taken:

- A new section S4 was added to the Supplementary Information, showing corroborative Hall measurements, including a new Figure S3.

S4: Low-temperature transport

Figure S3 shows electronic transport data obtained in a mimicked Hall bar geometry, for which an approximately $1 \times 5 \text{ mm}^2$ -sized stripe was cut from the sample and contacted by wire bonding. Figure S3a shows the temperature-dependent resistance of the samples, while Figure S3b shows the field-dependence of the transversal Hall resistance at various temperature. Below temperatures around 30 K a non-linear Hall resistance is observed, which can be modeled based on a two-band conduction mechanism, typically applied for LAO/STO interfaces.^{14,15} The resulting mobilities and carrier densities are displayed in Figures S3c and S3d, revealing a high-mobility-low-density and a low-mobility-high-density electron species, with typical figures for LAO/STO samples. Note that in DC transport the absolute mobility values can differ strongly from mid-infrared mobilities, as explained in the main text.

Figure S3: Electronic transport data from mimicked Hall-bar geometry from a $1 \times 5 \text{ mm}^2$ stripe of the LAO/STO sample. **a** Temperature-dependent resistance and **b** field-dependence of the transversal Hall resistance at various temperatures. **c** The resulting mobilities and **d** carrier densities resulting from assuming a two-band conduction mechanism.

- A reference to this section was added to the main text: “Values for carrier concentration and mobility were picked in accordance with Hall measurements of the sample (cf. Supplementary Information S4), (...)”

- Point raised:** Parts of the description of the “fingerprint” region for particular layers would also benefit from further clarification. For example, the authors suggest that s-SNOM possesses both high surface sensitivity and sub-surface capabilities, which appears contradictory, and would benefit from greater discussion of when either regime would be probed.

Answer: The seeming contradiction arises from the exponential decay of s-SNOM near-fields, with a decay length of several ten nanometers into the sample. To make this more intuitive, a direct comparison with other methods can be helpful: s-SNOM has subsurface capabilities compared to AFM or STM, as the near-fields penetrate into the sample and gather information from the depth below the immediate surface, however, s-SNOM is extremely surface sensitive compared to far-field optical measurements, which average over several μm depth for many materials in the mid-infrared spectral range. To address this in the manuscript, we now explain the origin of the surface sensitivity and the sub-surface capabilities in more detail.

Action taken:

- The introduction in the main text was expanded and now reads:

The near-fields of s-SNOM decay exponentially with distance to the tip, in the order of several ten nanometers^{25–27}, thus even layers of few nanometers thickness can significantly influence the scattering signal,^{28–30} especially in the vicinity of zero-crossings of the materials' $\text{Re}[\varepsilon]$.^{23,31,32} The probing depth depends also on the experimental parameters³³, and can thus be adjusted between several ten and hundred nanometers. For our layered system, this leads to a combined scattering response of the STO substrate, the LAO top layer, and the 2DEG in between, dependent on their respective permittivities.

Reviewer 2:

Manuscript entitled “Mid-infrared near-field fingerprint spectroscopy of the 2D electron gas in LaAlO₃/SrTiO₃ at low temperatures” by Julian Barnett et. al., presents a detailed study on exploiting the theoretical predictions of a characteristic spectral region for the s-SNOM investigation of the LAO/STO 2DEG at low temperatures. In summary, the experimental data are of high quality and well backed-up by supporting information. I find this paper to be very-well conceived and nicely written. I recommend publication in Nature Communication after some changes.

1. **Point raised:** In Figures 2a and b, the authors have shown the variation in the intensity of peaks depending on the mobility and carrier density. But the physical explanation is lacking in the manuscript. Also the reasons why the mobility change only in the first peak(I), but the carrier density change in all three peaks (I,II,III). It would be important to include a hypothesis (possible origins) in a qualitative way.

Answer: We thank the Reviewer for the high praise on the quality of our experimental data and Supplementary Information. We agree that the physical explanation of the origin of the peaks in near-field fingerprint spectroscopy was lacking and have now severely expanded on this. For this, we added a new section S2 in the Supplementary Information, that deals with the physical origin of the near-field fingerprint spectrum, and linked it at different positions in the main text. Additionally, we included the new example of doped InAs including a detailed discussion into the main text (cf. response to Reviewer 1).

Action taken:

- A new Figure 4 was added to the manuscript, complete with description in the main text (see reproduced text in the answer to Reviewer 1)
- A new section S2 was added to the Supplementary Information, complete with new Figure S2:

S2: Generalized near-field fingerprint spectroscopy

The findings on near-field fingerprint spectroscopy presented in the main text are an advanced application of s-SNOM, that is new in the reported frequency range. The working principle of s-SNOM is based on coupled electromagnetic near-fields between tip and sample at optical frequencies. Assuming constant tip properties and tip-sample distance, the scattering signal is mostly dependent on the dielectric properties of the sample, introducing a frequency dependence via the dielectric function $\varepsilon(\nu)$. Figure S2a shows a generalized coupling function of the Au-normalized scattering amplitude s_2/s_2^{Au} , which depends on the real (x -axis) and imaginary part (grey scale) of ε . While s_2/s_2^{Au} converges for highly positive and negative values of $\text{Re}[\varepsilon]$, different behavior can be observed

between values of +20 and -20. For low imaginary part (light grey curve), a high scattering amplitude is visible at slightly negative $\text{Re}[\varepsilon]$ (“near-field resonance”), while a minimum in scattering amplitude occurs at $\text{Re}[\varepsilon] \approx 1$. Thus, changes to ε of the sample result in non-linear changes of the scattering amplitude, with the highest sensitivity (maximum slope) usually found around $\text{Re}[\varepsilon] \approx -1$. This can be seen in the zoom-in in Figure S2b, where the range of maximum slope is highlighted in red. Decreasing $\text{Re}[\varepsilon]$ by a small amount (blue-green arrow), e. g. by introducing free charge carriers to the material, leads to a change in s_2/s_2^{Au} that is strongest in this range.

Figure S2c shows the dielectric function of a bulk STO single crystal (blue) in the range of the highest-frequency phonon mode. Additionally, a small shift of -0.3 to $\text{Re}[\varepsilon]$ is presented in green, much better visible in the zoom-in in Figure S2d, which shows the range around the LO frequency (zero-crossing) of the STO phonon. The red-shaded area again indicates the region of highest s-SNOM sensitivity (cf. Figure 4 of the main text), linked to Figure S2b via dashed lines. The resulting s-SNOM scattering amplitude is shown in Figure S2e, which can be directly compared to Figure 2f of the main text. In this case, s_2/s_2^{Au} is shown for bulk STO (blue curve) compared to the slightly decreased $\text{Re}[\varepsilon]$ (dashed green curve), resulting in changes I) and III) due to the perturbation, similar to the description of the conducting and insulating sample in the main text. However, change II) of Figure 1f is absent (greyed out), as it is a specific feature of the layered LAO/2DEG/STO stack. This shows that adding layers can lead to additional peaks in the fingerprint spectrum, which helps to disentangle different contributions to the s-SNOM signal.

Referencing the perturbed (changed ε) to the unperturbed case (unchanged ε) results in the normalized spectrum (solid green curve) shown in Figure S2f. Here, two peaks arise from the changes I) and III), that are characteristic for the perturbation to the dielectric function of the material. As these peaks are located in the frequency range where s_2/s_2^{Au} is most sensitive to changes of ε (red-shaded region), this near-field “fingerprint” is ideally suited to track slight variations of the optical properties of the material.

Figure S2: Working principle of fingerprint spectroscopy (simulations) for arbitrary perturbations $\Delta\epsilon$ of a bulk material. **a** Coupling function of s-SNOM, i. e. the dependence of the scattering amplitude s_2/s_2^{Au} on the sample dielectric function for the surface of a bulk material. For low imaginary part (light grey), a pronounced near-field resonance (peak) is visible at slightly negative values of $Re[\epsilon]$, which yields the highest absolute scattering signals. **b** Zoom-in of the coupling function, showing the influence of the slightly changed dielectric function on s_2/s_2^{Au} by shifting 0.3 to the left (blue-green arrow). The red shaded region indicates the range of $Re[\epsilon]$ where the change of the scattering signal is strongest (maximum slope). **c** Real (solid line) and imaginary part (dashed line) of the dielectric function of bulk STO (blue), showing the highest frequency phonon mode; a perturbation $\Delta\epsilon = -0.3$ (green) introduced to the real part is barely visible on this scale. **d** Zoom-in of the dielectric function close to the LO-frequency (zero-crossing), showcasing the shifted real part (blue-green arrow). **e** Resulting near-field amplitude spectrum s_2/s_2^{Au} of bulk STO (blue) compared to the result of the slightly changed $Re[\epsilon]$ (dashed green). Changes I) and III) are highlighted (cf. Figure 1f of the main text), while change II) is not present (greyed out). **f** The normalization procedure results in two characteristic peaks (cf. Figure 1g of the main text), with position, height and shape depending on the magnitude of the change as well as on the original scattering spectrum of the bulk material. The missing peak II) (grey label) is a feature of the layered system and thus not present in the bulk case shown here.

At this point, it should be emphasized that the behavior described here is universally applicable to any bulk or layered material that exhibits a zero-crossing of $Re[\epsilon]$ at low $Im[\epsilon]$, such as the LO frequency of a phonon mode. As was shown in the main text, adding layers can lead to the appearance of additional peaks, depending on the relative position of the zero-crossings of $Re[\epsilon]$ of each material. Furthermore, the perturbation $\Delta\epsilon$ as shown here typically results from adding free charge carriers, making this a model case for doped oxides, topological insulators and conducting 2D materials.

2. **Point raised:** While the variation of carrier density and mobility could result in clear variation based on simulation (Fig. 2a and b), it's difficult to link the spectra to the exact carrier density and mobility for an unknown system (LAO/STO is a known-answer question), especially when there are extrinsic contributions in play, such as defects, local inhomogeneous, etc. Therefore, one would have a feeling that other approaches (e.g., Hall measurement) are more reliable.

Answer: We have now added corroborative Hall measurements in the Supplementary Information (cf. response to Reviewer 1). For s-SNOM measurements of new materials, a step-by-step approach including reference samples and correlative methods, such as Hall measurements, are necessary to build a physical model and gain meaningful insight about material properties from the recorded scattering signals. However, it is very important to note that Hall measurements do not have the nanoscale resolution of s-SNOM and are therefore unable to resolve spatially varying electronic properties or defects.

Action taken:

- We added corroborative Hall measurements as a new section SI4 in the Supplementary Information, and reference to this in the main text (cf. response to Reviewer 1).
- To explain the importance of correlative measurements for s-SNOM, the following sentences were added to the manuscript:

(...) For this, it is important to note that a description of sample materials and geometry must be built into the theoretical model to understand the link between s-SNOM signal and material properties. As such, additional knowledge from correlative measurements is necessary to gain meaningful insight into the near-field fingerprint spectrum. The resulting pattern, however, is highly sensitive to small changes in optical properties resulting from local inhomogeneities, surface modifications, or deliberately influencing the material (e. g. via gating), making s-SNOM with near-field fingerprint spectroscopy a powerful tool for nanoscale material analysis.

3. **Point raised:** The results in Fig. 2c and d are not convincing. The mechanism for the shift and variation of peak II and III are not clear and the simulations probably should be optimized.

Answer: While the simulations are not a perfect match, we mostly focused on peak (I) and (II), due to the general difficulty of describing the third peak correctly. We have now added possible physical explanations for the deviation of peak (III) between simulation and measurement to the main text and made the focus of the subsequent discussion clearer. Additionally, the new section S2 in the Supplementary Information gives additional insight into the mechanism behind the near-field fingerprint spectrum (cf. response to point 1)

Action taken:

- The main text was modified to include possible mechanisms for the broader peak (III) in measurements compared to simulations and the goal of the parameter variation was made clearer:

Figure 2c presents the first ever s-SNOM measurements of the 2DEG fingerprint region, for two different positions (black, grey) on LAO/STO with 2DEG (cf. Methods and Supplementary information for details). For the first position (black curve), a high peak (I) around 750 cm^{-1} , a small peak (II) at 790 cm^{-1} and a broad peak (III) around 850 cm^{-1} can be observed, which generally fits well to the theoretical predictions presented in Figures 2a/b, especially for peak (I) and (II). In direct comparison, peak (III) seems much broader in the measurement, which might result from frequency-dependent

damping $\gamma(\omega)$ for the free charge carriers around the LO-frequency of the STO substrate⁴⁶, which is not yet well understood and not included in the model. The higher background at 900 cm^{-1} in both measurements compared to the respective simulations could be explained by a change in the high-frequency limit ϵ_∞ of the dielectric function. Such a change in ϵ_∞ was shown to influence STO near-field signals upon doping,^{41,42} possibly also contributing to the broader peak (III).

For the second parameter set (grey), the goal was to reproduce the main characteristics of this near-field fingerprint spectrum, i.e. the different intensity distribution between the first two peaks (I) and (II), as well as the shift of the minimum between these two peaks to lower frequencies. However, this matching was not possible by changing the electronic properties alone. Instead, small changes to the LAO/STO phonon properties were necessary to achieve the frequency shift of the minimum (cf. Supplementary Information S3) and the third maximum (III). This points towards inhomogeneities of the LAO layer, possibly due to differently strained areas after cooling, as strain can lead to significant shift of the phonon frequencies in oxides.⁴³ The inhomogeneous 2DEG properties also detected by this method could result from magnetic or structural domains observed for LAO/STO 2DEGs at low temperature.^{44–47}

4. **Point raised:** It's not mandatory, but it's suggested to remove Fig. s1 a & b, and put Fig. s1c and d as insets in Fig. 2 a and b, respectively.

Answer: We agree that this is a more elegant solution to present the data and implemented it as proposed.

Action taken:

- Figure 3 was changed to incorporate the insets showing the maximum height of peak (I), including a change to the caption and a mention in the main text:

- The caption now reads: “a Simulation for 2DEG mobilities μ between 5 cm^2/Vs (dark red) and 80 cm^2/Vs (yellow), with a constant 2DEG carrier concentration $n_{2D} = 4 \cdot 10^{13} \text{ cm}^{-2}$. The inset shows the maximum value of peak I) with rising mobility. b Simulation for different n_{2D}

between $2 \cdot 10^{13} \text{ cm}^{-2}$ (dark red) and $6 \cdot 10^{13} \text{ cm}^{-2}$ (yellow), with a constant $\mu = 20 \text{ cm}^2/\text{Vs}$. The inset shows the maximum value of peak I) with rising carrier concentration.”

- The main text now reads: “Additionally, the scaling behavior is very different, as can be seen from the insets in both plots, which show the maximum height of peak I) with rising mobility or carrier concentration, respectively. Even though μ is doubled between each curve in Figure 2a, the increase in peak height diminishes with each step and saturation is reached around $80 \text{ cm}^2/\text{Vs}$ (yellow curve). (...)”

Reviewer 3:

In the present manuscript, the authors show mid-infrared near-field fingerprint spectroscopy of the two-dimensional electron gas (2DES) at the LaAlO₃/SrTiO₃ (LAO/STO) interface at 8 K, and further obtain the carrier density and mobility by theoretical fitting.

1. **Point raised:** The results are interesting, but do not provide any new insight into the nature of the 2DES at the LAO/STO interface. Actually, the LAO/STO 2DES have been extensively studied, especially by electronic transport, to quantitatively reveal its nature. Characterizations using near-field spectroscopy similar to the present work have been performed for 2DES at oxide interfaces (include LAO/STO), see Nat. Commun. 10, 2774 (2019); Adv. Funct. Mater. 30, 2004767 (2020); Nano Lett. 22, 2755 (2022). These studies have also obtained successfully the carrier density and mobility, though the adopted wavelength range may be different from the present study. I fail to see any brand-new insight from the present study.

Answer: In the original version, we weakened our own claim by respectfully glossing over weaknesses of previous papers. We rectified this in the new version by being more precise in describing the knowledge gap that we address with our manuscript. While indications of different influence for carrier concentration could be found, previous publications did not achieve this goal. Instead,

- in (Nat. Commun. 10, 2774 (2019)) an overfitting of only 3 datapoints was used, with many different combinations of carrier concentration and mobility yielding similar results,
- in Adv. Funct. Mater. 30, 2004767 (2020), the mobility was set to a fixed value to get a qualitative influence of the carrier concentration, and
- for the SrTiO₃/TiO₂ interface in Nano Lett. 22, 2755 (2022), very similar broadband spectra to the phonon near-field resonances of LAO/STO in Adv. Funct. Mater. 30, 2004767 (2020) were recorded in a cross-section geometry. Again, the mobility was set to a fixed value to investigate the spatial variation of the carrier concentration.

The mutual influence of carrier concentration and mobility is covered in detail in Nanoscale Adv., 2021, 3, 4145-4155, where a detailed discussion on the difficulty of separating both properties is included in the Supporting Information. There it is shown that a variation in carrier concentration can be compensated by a variation in mobility to a large degree, resulting in degenerate spectra within the frequency ranges previously available for experiments. The point of the near-field fingerprint spectroscopy method described in this manuscript is to generate enough characteristic features that fitting procedures yield a more meaningful result, which cannot be reproduced by many different parameter sets.

Action taken:

- When describing previous literature, we do not claim that the extraction of local electronic properties was already sufficiently achieved and emphasize the problem with compensating influences of carrier concentration and mobility:

(...) s-SNOM was shown to be sensitive to the 2DEG in LAO/STO²¹ and even allow in principle for the extraction of local electronic properties with nanoscale lateral resolution.

(...) During these studies, direct mapping of the local charge carrier density turned out to be difficult, as the influence of different parameters, e.g. carrier concentration and mobility, on near-field spectra can compensate, resulting in similar near-field spectra.²⁴

(...) Grey-shaded areas indicate spectral regions where the 2DEG was experimentally investigated in previous studies. These studies were limited to the influence of the 2DEG on i) the phonon near-field resonance of STO ($< 750 \text{ cm}^{-1}$), where the s-SNOM signal is generally high,²³ or ii) the off-resonance scattering response in the spectral window of light sources with a high signal-to-noise ratio, such as CO₂ lasers ($> 920 \text{ cm}^{-1}$).^{22,24} In both regions, the general influence of free charge carriers on near-field spectra is only indirect, either i) as an additional source of damping (increased $\text{Im}[\varepsilon]$) or ii) as a constant background (increased high-frequency limit ε_∞ , cf. Supplementary Information S3). Thus, a clear separation of carrier concentration and mobility was not previously possible.

(...) Note that this fingerprint spectrum is constrained to the shown frequency range by the phonon properties of the two materials and could not be experimentally investigated in previous publications due to lack of light sources with sufficient signal-to-noise ratio (cf. Figure S1 in the Supplementary Information).

- We added the following sentence to the description of our results:

Thus, the characteristic peaks of the near-field fingerprint spectrum are predicted to behave differently for the two parameters, finally allowing for direct experimental access to separated \$n\$ and \$\mu\$.

- We also added a new section S1 to the Supplementary Information, including a new Figure S1, which compares the new signal quality to that of our previous publication (Adv. Funct. Mater. 30, 2004767 (2020)), showing the vast improvement brought by the new light source, even at much higher measurement speeds:

S1: Comparison to previously published results

The main text discusses that previous publications of s-SNOM on the LAO/STO 2DEG were limited by the availability of light sources. Two cases were differentiated in Figure 1d: a) to investigate at higher frequencies ($> 930 \text{ cm}^{-1}$),¹ where high-intensity light sources such as CO₂-laser or QCLs were available but the sensitivity of s_2/s_2^{Au} to changes of the dielectric function is much lower, or b) to investigate the phonon near-field resonance of STO,² where the scattering efficiency is higher but the influence of the 2DEG is mostly visible as additional damping via its contributions to the imaginary part of the dielectric function. Near-field fingerprint spectroscopy via self-referencing, as presented in this paper, allows for a much higher sensitivity of the s-SNOM signal to the electronic properties of the 2DEG. However, the scattering efficiency (absolute scattering amplitude) at the positions of the peaks is very low. As a result, the peaks will be lost if signal-to-noise of the light source is too low. To visualize this, published results from Synchrotron measurements of the same samples are compared directly with the new measurements shown in this publication.

Figure S1a shows the absolute scattering amplitude s_2 of the conducting LAO/STO sample, measured with a commercial s-SNOM (Neaspec GmbH) in nanoFTIR setup with synchrotron illumination from the Metrology Light Source (MLS) at Physikalisch-Technische Bundesanstalt (PTB) Berlin.³ The storage ring

was operated in a mode characterized by a low horizontal emittance, therefore leading to both a low beam size and a low beam divergence, making this mode particularly suited for s-SNOM measurements, due to high illumination power at the tip apex.⁴ The black and red curve show two consecutive measurements at the same position, indicating the reproducibility of the measurement. In the spectral range between 700 and 750 cm^{-1} , the right flank of the near-field resonance peak is visible, whereas the signal at higher frequencies (above 750 cm^{-1}) is very low, as expected from theory (cf. Figure S2). Zooming in by a factor of 10 on the vertical axis (Figure S1b) shows that the overall reproducibility above 750 cm^{-1} is poor due to low signal-to-noise, which does not allow for self-referencing in this spectral region, as dividing by small numbers with high variation leads to a near-field fingerprint spectrum that consists of erroneous peaks.

Figures S1c and S1d present comparable measurement data used in this publication (cf. Figure 2 of main text), related to the region of highest sensitivity $\delta s_n / \delta \epsilon$ (red-shaded area, cf. Figure S2). The reproducibility of individual measurements is much better, which results in reproducible normalized spectra, as shown in the main text. Thus, the availability of new light sources (cf. Table 1) allows for the previously impossible utilization of near-field fingerprint spectroscopy to investigate small perturbations to the dielectric function in the vicinity of zero-crossings of $\text{Re}[\epsilon]$.

Figure S1: Comparison of measurements to previously published results. **a** Fourier-transformed scattering amplitude s_2 of two consecutive nanoFTIR measurements with Synchrotron illumination (see text for details), used in the previous publication.² **b** Zoom-in by a factor of $\times 10$ on the vertical axis. **c** Scattering amplitude s_2 of two consecutive s-SNOM measurements with Laser illumination (see main text for details). The measurement speed was increased by approximately a factor of $\times 10$ compared to measurements in Subfigure S1a. **d** Zoom-in by a factor of $\times 10$ on the vertical axis. The red-shaded area denotes the previously identified region of highest sensitivity $\delta s_n / \delta \epsilon$ (cf. Figure S2).

Table 1: Rough comparison of typical light sources used in SNOM.⁶⁻⁸

light source	spectral width at one setting [cm^{-1}]	Overall tuning range [cm^{-1}]	est. spectral irradiance at focus $\text{W}/\text{cm}^2/\text{cm}^{-1}$
thermal source	4000	-	9×10^{-4}
very broadband laser	400	700-2500	$0.1-2 \times 10^{-2}$

synchrotron	2000	-	
broadband laser	75	500-2200	2-37
tunable laser (this paper)	3	625-2000	1,100
QCL (cw)	1	850-2800	17,000

2. **Point raised:** The authors also demonstrate the lateral inhomogeneity by near-field imaging, which has been well established previously. And the near-field results in this study do not help to reveal the origin of the inhomogeneity, at least the authors do not discuss about. (...) The hardcore experimental data are very limited, and the reproducibility is poor. For example, the two images scanned at the same area dramatically different, as shown in Fig. S2. The authors attribute such nonreproducibility to the changes in the local potential, which is not very convincing to me.

Answer: We agree with the referee, that imaging of lateral inhomogeneities of the LAO/STO 2DEG with s-SNOM has been done before, for example in our paper [Rose 2021, *Nanoscale Adv.* 3(14)], where we found a change to the 2DEG properties close to step edges, which we attributed to a change of substrate termination. However, at that time it was not possible to differentiate between a change in carrier concentration and a change in mobility, which is relevant to understand the physical mechanism of the inhomogeneity. With our new method of fingerprint spectroscopy such a distinction is now possible, and will be applied to various types of inhomogeneities (e.g. termination, step edges, LAO thickness, defects, light induced properties...) in the future.

The fabrication process of our investigated sample was specifically designed for homogeneous 2DEG properties, to get a fundamental understanding of LAO/STO s-SNOM spectra at low temperatures. Nevertheless, we surprisingly found inhomogeneities in the recorded s-SNOM images (Figure 3) and used them as a benchmark for the subwavelength resolution of our new method. We show that the local variation can be explained by a change of carrier mobility, as opposed to a change of carrier concentration. The changed mobility itself could arise from a variety of different origins, which we were unable to determine within the boundary conditions of the available experiments.

Addressing the nonreproducibility of the images in Figure S2, it is discussed in literature that the 2DEG mobility can be influenced in low-temperature scanning probe measurements of LAO/STO, e.g. due to electrostatic gating from the irradiated AFM tip, persistent photoconductivity, or frozen condensates (see main text for citations). This is a topic of ongoing research and needs to be addressed in future measurements with specially designed sample layouts and experimental conditions.

However, compared to previous measurements, the data quality and reproducibility of our near-field spectra is much better (cf. new Figure S1 in the new section S1 of the Supplementary Information) due to technological advances. This is a huge methodological development, as it enables us to create normalized near-field fingerprint spectra, finally allowing for the separation of carrier concentration and mobility in s-SNOM investigations of the LAO/STO 2DEG.

We agree with the reviewer that the imaging data is limited, as imaging was not the focus of our investigation and specially designed sample layouts and experimental conditions would be necessary. However, as another reviewer stated, these cryo-SNOM experiments are extremely challenging and contain many unknowns, and the possibility to undertake these measurements was concluded and could not be repeated for now.

We updated the manuscript to be more explicit and to include citations for possible causes of the inhomogeneities observed in our s-SNOM images.

Action taken:

- The explanation of Figure 3 in the main text was changed to explain the origin of the inhomogeneity more clearly:

This behavior is very similar to Figure 2a and can be reproduced by a variation of the 2DEG mobility μ between 8 cm²/Vs and 40 cm²/Vs (Figure 3c), showing that with near-field fingerprint spectroscopy the local inhomogeneities observed in the s-SNOM image (Figure 3a) can be linked to a laterally varying 2DEG mobility. Additional s-SNOM images recorded after the line scan (cf. Supplementary Information S5) indicate that the inhomogeneous mobility could be affected by the previous measurement of the same area. An influence of low-temperature scanning probe measurements on the electron mobility of LAO/STO is known from literature and a current topic of research, e.g. arising due to electrostatic gating from the irradiated AFM tip,²² persistent photoconductivity,^{48,49} or frozen condensates.⁵⁰ Using the increased sensitivity of near-field fingerprint spectroscopy to the local electronic properties, the physical origin of such variations can now be investigated in the future.

- The explanation was also added to the Supplementary Information S5:

This could indicate that the preceding s-SNOM measurement itself is responsible for the observed change in near-field response. Similar effects in low-temperature scanning probe measurements are known from literature and a current topic of research, e.g. arising due to electrostatic gating from the irradiated AFM tip,²² persistent photoconductivity,^{49,50} or frozen condensates.⁵¹

3. **Point raised:** The authors adopt 700 cm⁻¹ for imaging (Fig. 3a), which is not any of the identified fingerprint peaks (750 cm⁻¹, 775 cm⁻¹, 860 cm⁻¹). Why? From Fig. 2, the intensity at 700 cm⁻¹ is not sensitive to the inhomogeneity at all.

Answer: We thank Reviewer 3 for this very good point! During the measurement of this image, we did not know a frequency for *good contrast* between different electronic properties and instead picked the frequency with the *highest overall signal* (cf. Figure 1e), close to the phonon near-field resonance, to ensure sufficient signal-to-noise ratio to record good images with reasonable scan times. The 2D images were visible during the measurement, prompting us to do a detailed spectroscopic investigation of the inhomogeneity we found. At this time, near-field fingerprint spectra were not yet thought of. The idea to calculate them and investigate the results in more detail only arose later, during data evaluation. As stated in the previous answer, the possibility to undertake these challenging cryo-SNOM experiments was concluded and could unfortunately not be repeated for now. In the future, we aim to record images at frequencies that were identified to show the highest contrast and strongest changes in near-field fingerprint spectra.

Action taken:

- The following sentence was added to the main text:

To further investigate how these near-field fingerprint spectra (cf. Figure 2c) vary for different positions on the sample, we performed preliminary imaging (Figure 3a) at a frequency of 700 cm⁻¹, where the scattering signal is highest due to the strong phonon near-field response (cf. Figure 1d/e). These single frequency s-SNOM images show significant variations in amplitude s_2 (Figure 3a) and phase ϕ_2 (Supplementary Information S5).

- The following paragraph was added to section S5 of the Supplementary Information to make this clear:

For these images, an illumination frequency with the *highest overall signal* around the phonon near-field resonance was picked (cf. Figure 1e of the main text), to ensure sufficient signal-to-noise ratio for image recording. At this time, illumination frequencies for *high contrast* between different electronic properties were not yet known, as the fingerprint spectra were calculated afterwards.

4. **Point raised:** It is better to compare the experimental data (Fig. 2c) and fitting data (Fig. 2d) directly in the same figure. It seems to me the fitting is not really good, considering it is a multi-parameter fitting.

Answer: Generally, s-SNOM spectra are non-trivial and fitting requires a physical model with many parameters to gain meaningful insight about material properties from the recorded scattering signals. In Figure 2, our goal was to show that the three theoretically predicted peaks resulting from the normalizing procedure of near-field fingerprint spectroscopy on the LAO/STO 2DEG can indeed be found in experiments. We also show that they differ in position, height and width in a similar manner as predicted, which allows conclusions on the properties of the 2DEG.

We found that the physical origin of the fingerprint peaks lies in the coupling function of s-SNOM and the influence of perturbations to the dielectric function caused by additional free charge carriers. We now generalize this for other material systems in our manuscript (see answers above), such as accumulation layers in doped InAs (new Figure 4) and doped STO (new Figure S2 in the Supplementary Information).

We should note, that the peaks can also be influenced by other minor perturbations of the dielectric function, e.g. due to a shift in ω_{LO} from strain in the LAO layer. As a result, a direct quantitative analysis would require even more fit parameters for these minor perturbations, which is beyond the scope of our work.

Action taken:

- A new Figure 4 was added to the manuscript, complete with description in the main text (see reproduced text in the answer to Reviewer 1)
- A new section S2 was added to the Supplementary Information, complete with new Figure S2 (see reproduced text in the answer to Reviewer 2)

Other changes:

- Small changes were introduced throughout the manuscript to improve the quality of writing and understanding for the reader; all changes are highlighted in blue in the manuscript.